# Random Rank: The One and Only Strategyproof and Proportionally Fair Randomized Facility Location Mechanism

**Haris Aziz**
Computer Science and Engineering
UNSW Sydney
Sydney NSW 2052, Australia
`haris.aziz@unsw.edu.au`

**Alexander Lam**
Computer Science and Engineering
UNSW Sydney
Sydney NSW 2052, Australia
`alexander.lam1@unsw.edu.au`

**Mashbat Suzuki**
Computer Science and Engineering
UNSW Sydney
Sydney NSW 2052, Australia
`mashbat.suzuki@unsw.edu.au`

**Toby Walsh**
Computer Science and Engineering
UNSW Sydney
Sydney NSW 2052, Australia
`t.walsh@unsw.edu.au`

## Abstract

Proportionality is an attractive fairness concept that has been applied to a range of problems including the facility location problem, a classic problem in social choice. In our work, we propose a concept called *Strong Proportionality*, which ensures that when there are two groups of agents at different locations, both groups incur the same total cost. We show that although Strong Proportionality is a well-motivated and basic axiom, there is no deterministic strategyproof mechanism satisfying the property. We then identify a randomized mechanism called Random Rank (which uniformly selects a number $k$ between 1 to $n$ and locates the facility at the $k$'th highest agent location) which satisfies Strong Proportionality in expectation. Our main theorem characterizes Random Rank as the unique mechanism that achieves universal truthfulness, universal anonymity, and Strong Proportionality in expectation among all randomized mechanisms. Finally, we show via the AverageOrRandomRank mechanism that even stronger ex-post fairness guarantees can be achieved by weakening universal truthfulness to strategyproofness in expectation.

## 1 Introduction

The facility location problem has been widely studied in various literatures. In the classic setting, we are given a set of agent locations on the unit interval, and are tasked with finding an 'ideal' location on the interval to place a facility. Each agent incurs a cost equal to its distance from the facility, and therefore wants the facility to be placed as close as possible to its own location. The problem applies to many real-world scenarios, such as the geographical placement of public facilities [Schummer and Vohra, 2002; Miyagawa, 2001], or the choice of a political representative [Moulin, 1980; Feldman *et al.*, 2016]. The problem can be similarly applied to aggregate votes on what proportion of a budget is dedicated to a certain project [Freeman *et al.*, 2021], and the 1D-metric can be extended to a graph to model router placement on a network [Hakimi, 1965].

In this paper, we explore the space of facility location mechanisms that have desirable fairness and strategic properties. In particular, a mechanism should satisfy anonymity, which requires that the outcome of the mechanism should not depend on the names of the agents. Secondly, we seek

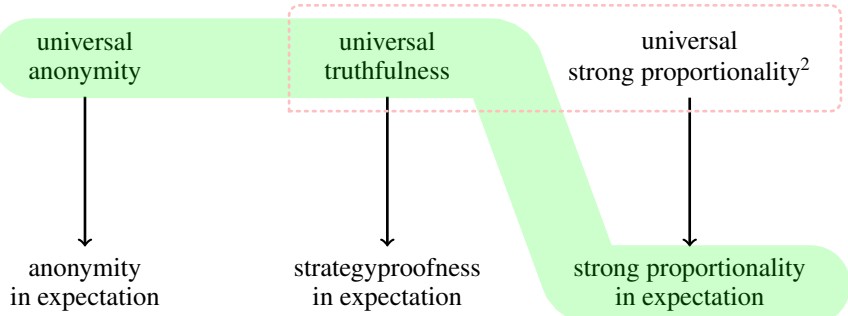

Figure 1: Logical relations between fairness and efficiency concepts. An arrow from (A) to (B) denotes that (A) implies (B). The properties in green are simultaneously satisfied by some algorithms. The properties in the dotted pink shapes are impossible to simultaneously satisfy.

mechanisms that are strategyproof and hence cannot be manipulated by agents misreporting their truthful private information (which in our case is the agents' location). Thirdly, we require that the mechanisms should be fair. We focus on a fairness concept called Strong Proportionality that has a deep foundation in the theory of fair collective decision making. It is based on the idea that when there are two groups of agents at different locations, both groups incur the same total cost. As a consequence, the facility is placed such that the ratio of distances between each group and the facility is inversely proportional to the ratio of their group sizes. In other words, the facility is placed "proportionally" closer to the larger group of agents. Our primary focus is on identifying mechanisms which satisfy all three properties discussed above.

In addition to the fairness axiom of Strong Proportionality, we focus on universal versions of truthfulness[1] and anonymity as desirable properties. These universal versions can be viewed as more robust variants of their counterpart axioms when considering randomized mechanisms. Universal truthful mechanisms randomize over deterministic strategyproof mechanisms, and do not incentivize an agent to misreport even when the random outcome is known (see e.g., [Nisan and Ronen, 2001] ). Similarly, universally anonymous mechanisms randomize over deterministic anonymous mechanisms, and are robust to an agent changing its labelling to achieve a better outcome even if they know the random outcome.

In our paper, we focus on randomized mechanisms as we show that no deterministic mechanism satisfies truthfulness along with Strong Proportionality. For this reason, we target Strong Proportionality *in expectation* and explore the space of mechanisms that satisfy universal anonymity, universal truthfulness and Strong Proportionality in expectation. Figure 1 describes the relationships between the fairness and efficiency concepts that we study.

**Contributions**   Our paper makes several contributions to the facility location literature in particular and the social choice literature in general. Firstly, we adopt the fairness axiom of Proportionality from the participatory budgeting literature and formulate the axiom of Strong Proportionality, which enforces the basic and natural requirement that when there are two groups of agents at different locations, the facility should be placed closer to the larger group of agents. We show that no deterministic and strategyproof mechanism can satisfy Strong Proportionality, and hence we turn to randomized mechanisms. Our main contribution is the characterization of the Random Rank mechanism as the unique mechanism satisfying universal truthfulness, universal anonymity and Strong Proportionality in expectation. When universal truthfulness is weakened to strategyproofness in expectation, we find that there is a family of mechanisms that achieve a fairer ex-post distribution of outcomes. Finally, we show that our main characterization still holds when Strong Proportionality is strengthened to Strong Proportional Fairness, a property which captures fairness guarantees for more general groups of agents with similar locations. Table 1 summarizes the properties satisfied by the mechanisms we discuss in the paper. Statements and results lacking proofs are proven in the appendix.

---

[1]We use truthfulness and strategyproofness interchangeably.

Table 1: Summary of properties satisfied by the mechanisms we study. We show that in the space of all randomized mechanisms, only the Random Rank mechanism satisfies each property specified below. *The AverageOrRR$-p$ mechanism is strategyproof in expectation if and only if $p \in [0, \frac{1}{2}]$.

| Mechanism | Universal Truthfulness | Strategyproofness in expectation | Universal Anonymity | Proportionality in expectation | Strong Proportionality in expectation |
|---|---|---|---|---|---|
| **Random Rank** | Yes | Yes | Yes | Yes | Yes |
| **Random Dictatorship** | Yes | Yes | No | Yes | Yes |
| **Random Phantom** | Yes | Yes | Yes | Yes | No |
| **AverageOrRR$-p$** | No | Yes* | Yes | Yes | Yes |
| **Median** | Yes | Yes | Yes | No | No |
| **Uniform Phantom** | Yes | Yes | Yes | Yes | No |

## 2 Related Work

**Randomized social choice functions.** Randomized schemes have been proposed for a variety of social choice contexts to circumvent impossibility results (e.g. see [Brandt, 2017; Aziz, 2019]). For instance, the famous impossibility result of the Gibbard-Sattherwaite theorem [Gibbard, 1973; Satterthwaite, 1975] can be overcome by using a random dictatorship that chooses an outcome uniformly at random, which was first studied by Gibbard [1977, 1978]. The random dictatorship is similar to the 'Random Rank' mechanism we propose in our main characterization result. A generalization of random dictatorship is applied to the $k-$facility location problem by Fotakis and Tzamos [2010], achieving strategyproofness in expectation and a constant approximation for optimal total cost. Other recent studies on randomized mechanisms in the facility location problem have investigated the tradeoff between a mechanism's approximation ratio and its variance [Procaccia *et al.*, 2018], and proposed strategyproof mechanisms which minimize the maximum agent envy [Cai *et al.*, 2016]. Finally, we note that our results differ from the existing characterizations of randomized facility location mechanisms by Ehlers *et al.* [2002] and Peters *et al.* [2014], as their definitions of strategyproofness revolve around stochastic dominance, whilst our main characterization uses stronger axiom of universal truthfulness. Furthermore, the mechanisms we discuss are additionally constrained to be proportionally fair and thus our characterizations are more succinct.

**Facility location problems.** Facility location problems have been widely studied in the fields of computer science, economics and operations research. We take a mechanism design approach, finding mechanisms which do not incentivize agents to misreport their locations. The origins of this approach can be traced to the seminal work by Moulin [1980], which provides a characterization of strategyproof mechanisms when agent preferences are single-peaked. He also provides further characterizations when the axioms of anonymity and Pareto efficiency are imposed. There have since been many extensions of facility location mechanism design that build off these results, such as the placement of multiple facilities by Miyagawa [1998, 2001] and Fotakis and Tzamos [2013], as well as when agents have fractional or optional preferences [Fong *et al.*, 2018; Chen *et al.*, 2020]. For further related work, we refer the reader to a survey by Chan *et al.* [2021]. A connection between voting and facility location problems is drawn by Feldman *et al.* [2016], in which mechanisms minimizing social cost are formulated for the problem where there is a fixed, limited set of candidates on the real line.

**Fairness in collective decision problems.** Fairness constraints and measures have been formulated in a wide variety of different collective decision problems [Steinhaus, 1948; Dummett, 1997; Aziz *et al.*, 2019]. Our paper draws from the area of participatory budgeting, in which various axioms have been proposed to represent proportional fairness concerns for groups of agents with similar preferences [Aziz *et al.*, 2019; Pierczyński *et al.*, 2021]. Specifically, the axiom of "Proportionality" we use is drawn from the work on participatory budgeting by Freeman *et al.* [2021], and our new axiom of "Strong Proportionality" is, as the name suggests, a stronger version of the axiom. Our paper is most closely related to the work by Aziz *et al.* [2021], providing a characterization of strategyproof and proportionally fair mechanisms, but in a much more general randomized setting which introduces new conceptual and technical challenges as we see in later sections. Another related paper by Zhou *et al.* [2022] has similar underlying motivations, exploring group fairness cost objectives and proposing strategyproof mechanisms that approximate these objectives. We note that the notions of proportional

---

[2]A randomized mechanism satisfies *universal Strong Proportionality* if it is a distribution over deterministic mechanisms satisfying Strong Proportionality.

fairness explored in our paper differ from the fairness objective of egalitarian welfare explored widely in the facility location literature. For a comprehensive comparison between proportional and egalitarian fairness in the facility location problem, we refer the reader to the discussion by Aziz *et al.* [2021].

## 3   Preliminaries

Let $N = \{1, \ldots, n\}$ be a set of agents with $n \geq 2$, and let $X := [0, 1]$ be the domain of locations. The unit interval can be scaled and shifted to represent any closed and bounded interval on $\mathbb{R}$. As we will explain, our characterization results hold even when the domain is the real line. However, we restrict to $X := [0, 1]$ to remain consistent with the related literature [Massó and Moreno De Barreda, 2011; Aziz *et al.*, 2020, 2021; Freeman *et al.*, 2021]. Agent $i$'s location is denoted by $x_i \in X$; the profile of agent locations is denoted by $x = (x_1, \ldots, x_n) \in X^n$.

A *deterministic mechanism* is a mapping $f : X^n \to X$ from the agent location profile to a facility location. Given a location profile $x \in X^n$, agent $i$'s cost is $d(x_i, f(x)) := |x_i - f(x)|$.

A *randomized mechanism* is a probability distribution over deterministic mechanisms.[3]

A common requirement in mechanism design is that the mechanism's output should not depend on the agents' labels. That is, the mechanism outcome is invariant under a permutation of agent labels. Formally:

**Definition 1** (Anonymous). *A mechanism $f$ is* anonymous *if, for every location profile $x \in X^n$ and every bijection $\sigma : N \to N$,*
$$f(x_\sigma) = f(x),$$
*where $x_\sigma := (x_{\sigma(1)}, x_{\sigma(2)}, \ldots, x_{\sigma(n)})$.*

When considering randomized mechanisms, the notion of anonymity extends as follows:

- A randomized mechanism is *universally anonymous* if it is a distribution over deterministic anonymous mechanisms. If a mechanism is not universally anonymous, an agent that knows the random outcome may change its labelling to achieve a better outcome.

- A randomized mechanism is *anonymous in expectation* if the expected facility location does not change when the agents are relabelled.

Another normative property we seek in our paper is *truthfulness*, which incentivizes agents to report their locations truthfully.

**Definition 2** (Strategyproof). *A mechanism $f$ is* strategyproof *if, for every agent $i \in N$, we have for every $x_i'$ and for every $\hat{x}_{-i}$,[4]*
$$d(x_i, f(x_i, \hat{x}_{-i})) \leq d(x_i, f(x_i', \hat{x}_{-i})).$$

In the context of randomized mechanisms, the following are the two main notions of strategyproofness (truthfulness):

- A randomized mechanism is *universally truthful* if it is a distribution over deterministic strategyproof mechanisms. Under a universally truthful mechanism, an agent will not have an incentive to misreport its location even if it knows the random outcome.

- A randomized mechanism is *truthful in expectation* if no agent can improve its expected distance from the facility by misreporting its own location.

Given a location profile $x$, a facility location $y$ is said to be *Pareto optimal* if there exists no other facility location $y'$ such that $d(x_i, y') \leq d(x_i, y)$ for all $i$, and $d(x_i, y') < d(x_i, y)$ for at least one agent. A mechanism $f$ is said to be *(Pareto) efficient* if, for every location profile $x$, the facility location $f(x)$ is Pareto optimal. In our setting, Pareto optimality is equivalent to requiring that $f(x) \in [\min_{i \in N} x_i, \max_{i \in N} x_i]$.

A mechanism is *ex-post efficient* if it only gives positive support to Pareto efficient deterministic mechanisms.

---

[3]This definition of randomized mechanisms is commonly used in the literature [Dobzinski and Dughmi, 2013; Assadi and Singla, 2019]

[4]Here $x_{-i}$ denotes the location profile of all agents except $i$.

**Deterministic Anonymous and Strategyproof Mechanisms**

When agents have single-peaked preferences on the real line, Moulin [1980] has characterized the set of anonymous, strategyproof deterministic mechanisms as the class of 'Phantom' mechanisms, which place the facility at the median of the $n$ reported agent locations and $n + 1$ fixed 'phantom' locations. A similar characterization for the unit interval with symmetric single peaked preferences has been shown by Massó and Moreno De Barreda [2011].

**Definition 3** (Phantom Mechanism). *Given $n + 1$ fixed real numbers $0 \leq y_1 \leq \cdots \leq y_{n+1} \leq 1$ and a location profile $x$, a Phantom mechanism $f$ locates the facility at*

$$f(x) = \text{med}(x_1, \ldots, x_n, y_1, \ldots, y_{n+1}).$$

**Theorem 1** (Massó and Moreno De Barreda [2011]). *A (deterministic) facility location mechanism is anonymous and strategyproof if and only if it is a Phantom mechanism.*

Setting two of the phantoms at $0$ and $1$ also provides a characterization of anonymous, strategyproof and efficient mechanisms.

**Theorem 2** (Massó and Moreno De Barreda [2011]). *A (deterministic) facility location mechanism is anonymous, strategyproof and efficient if and only if it is a Phantom mechanism with phantoms $y_1 = 0$ and $y_{n+1} = 1$.*

Such a mechanism can also be interpreted as a Phantom mechanism with $n - 1$ phantom locations.

## 4 Proportional Fairness Axioms

In this section, we describe and motivate our axioms of proportional fairness used throughout the paper. We first state the fairness property of Proportionality which was defined in the context of the facility location problem by Aziz *et al.* [2021].

**Definition 4** (Proportionality). *A mechanism $f$ satisfies* Proportionality *if for any location profile $x \in \{0, 1\}^n$ and set $S$ of agents at the same location,*

$$d(x_i, f(x)) \leq \frac{n - |S|}{n} \quad \forall i \in S.$$

**Remark 1.** *Proportionality is equivalent to requiring that if all agents are located at the interval endpoints, the facility is placed at the average of the agents' locations.*

We consider a strengthening of Proportionality to allow for more general location profiles.[5]

**Definition 5** (Strong Proportionality). *A mechanism $f$ satisfies* Strong Proportionality *if for any location profile $x \in \{\alpha, \beta\}^n$ with $0 \leq \alpha < \beta \leq 1$ and set $S$ of agents at the same location,*

$$d(x_i, f(x)) \leq \frac{n - |S|}{n}(\beta - \alpha) \quad \forall i \in S.$$

**Remark 2.** Strong Proportionality *is equivalent to requiring that if agents are at most two different locations, the facility is placed at the average of the agents' locations.*

Strong Proportionality captures a number of basic fairness properties.

- Unanimity: If all agents are unanimous in their reported locations, then the facility is located at this same location.
- Equitable treatment of groups: When there are at most two groups of agents at different locations, both groups incur the same total cost.

While Strong Proportionality is a very basic and modest property, applying only to a small subset of possible location profiles, it cannot be satisfied by a deterministic, strategyproof mechanism.

**Proposition 1.** *No deterministic and strategyproof mechanism satisfies Strong Proportionality.*

---

[5]Strong Proportionality is applicable to settings lacking information on the endpoints of the domain, or when the domain has no endpoints (i.e. the interval is unbounded).

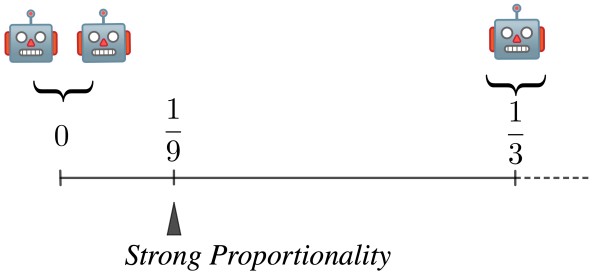

*Strong Proportionality*

Figure 2: Facility location problem with $n = 3$ agents and location profile $x = (0, 0, \frac{1}{3})$. For this profile, Strong Proportionality requires that the facility is placed at location $\frac{1}{9}$, which is achieved in expectation by the Random Rank mechanism.

As a result, we turn to the larger, more general space of randomized mechanisms. However, Proposition 1 implies no universally truthful mechanism can guarantee an output that achieves Strong Proportionality. Consequently, we attempt to find universally truthful mechanisms that instead achieve Strong Proportionality *in expectation*.

When considering randomized mechanisms, our proportional fairness axioms are adapted as follows:

**Definition 6** (Proportionality in expectation). *A mechanism $f$ satisfies* Proportionality in expectation *if for any location profile $x \in \{0, 1\}^n$ and set $S$ of agents at the same location,*

$$\mathbb{E}[d(x_i, f(x))] \leq \frac{n - |S|}{n} \quad \forall i \in S.$$

**Definition 7** (Strong Proportionality in expectation). *A mechanism $f$ satisfies* Strong Proportionality in expectation *if for any location profile $x \in \{\alpha, \beta\}^n$ with $0 \leq \alpha < \beta \leq 1$ and set $S$ of agents at the same location,*

$$\mathbb{E}[d(x_i, f(x))] \leq \frac{n - |S|}{n}(\beta - \alpha) \quad \forall i \in S.$$

## 5 Characterization of Universal Truthfulness and Strong Proportionality

In this section, we present the main result of the paper: a unique characterization of facility location mechanisms satisfying universal truthfulness, universal anonymity and Strong Proportionality in expectation. First, we define a mechanism which satisfies the aforementioned properties.

**Definition 8.** *The* Random Rank *mechanism $f_{RR}$ chooses a number $k$ at uniformly at random from $\{1, \ldots, n\}$ and outputs* $\mathrm{rank}^k(x) := \mathrm{med}(\underbrace{0, \ldots, 0}_{n-k}, \underbrace{1, \ldots, 1}_{k-1}, x_1, \ldots, x_n).$

**Remark 3.** *The Random Rank mechanism can also be interpreted as simply choosing a number $k$ at uniformly at random from $\{1, \ldots, n\}$ and outputting the $k$th largest agent location.*

As we will show, Random Rank is the only mechanism in the space of randomized mechanisms to satisfy universal truthfulness, universal anonymity and Strong Proportionality in expectation.

**Theorem 3.** *A mechanism is universally anonymous, universally truthful and Strong Proportional in expectation if and only if it is the Random Rank mechanism.*

*Proof.* ( $\Longleftarrow$ ) The Random Rank mechanism is universally anonymous and universally truthful as each realization of the mechanism, $\mathrm{rank}^k$, is strategyproof and anonymous by Theorem 2. Consider any location profile $x \in \{\alpha, \beta\}^n$, denote $S_\alpha$ as the set of agents located at $\alpha$ and $S_\beta = N \setminus S_\alpha$ as the set of agents located at $\beta$. For all $i \in S_\alpha$, we have $\mathbb{E}[d(x_i, f_{RR}(x))] = \frac{|S_\beta|}{n}(\beta - \alpha) = \frac{n - |S_\alpha|}{n}(\beta - \alpha)$ since Random Rank places the facility at $\alpha$ with probability $\frac{|S_\alpha|}{n}$, and at $\beta$ with probability $\frac{|S_\beta|}{n}$. By a similar and symmetric argument, for all $j \in S_\beta$, we have $\mathbb{E}[d(x_j, f_{RR}(x))] = \frac{n - |S_\beta|}{n}(\beta - \alpha)$. Thus Random Rank satisfies Strong Proportionality in expectation along with universal anonymity and universal truthfulness.

( $\implies$ ) Let $f$ be a mechanism satisfying universal anonymity, universal truthfulness, and Strong Proportionality in expectation. By Lemma 1 (which we will show later), $f$ also satisfies ex-post efficiency as Strong Proportionality in expectation implies Proportionality in expectation. Hence $f$ is a distribution over deterministic truthful, anonymous and efficient mechanisms. Furthermore, by Theorem 2, each of the deterministic mechanisms are Phantom mechanisms with $n-1$ phantom locations. Consequently we see that $f$ can be written as

$$f(x) = \text{med}(Y_1, \cdots, Y_{n-1}, x_1, \cdots, x_n),$$

where the $Y_i$'s are random variables corresponding to the locations of the $n-1$ Phantoms. Denote $Y_{(i)}$ as the random variable corresponding to the $i$'th order statistic, defined by sorting the values of $Y_1, \cdots Y_{n-1}$ in increasing order. We first show that the order statistics must satisfy a very restrictive set of properties.

**Claim 1:** $\Pr[Y_{(i)} = 1] = \frac{i}{n}$ and $\Pr[Y_{(i)} = 0] = \frac{n-i}{n}$ for each $i \in \{1, \cdots, n-1\}$.

*Proof of Claim 1.* Fix $i \in \{1, \cdots, n\}$. Consider the following family of location profiles

$$x^{\alpha,\beta} = (\underbrace{\alpha, \cdots, \alpha}_{n-i}, \underbrace{\beta, \cdots, \beta}_{i}) \qquad \alpha < \beta.$$

Let $S = \{1, \cdots, n-i\}$. Due to Strong Proportionality, we have

$$\mathbb{E}[d(x_1, f(x^{\alpha,\beta}))] \leq \frac{n-(n-i)}{n}(\beta-\alpha)$$
$$= \frac{i}{n}(\beta-\alpha).$$

On the other hand, we see that

$$\mathbb{E}[d(x_1, f(x^{\alpha,\beta}))] \geq 0 \cdot \Pr[Y_{(i)} \leq \alpha] + (\beta-\alpha)\Pr[Y_{(i)} \geq \beta]$$
$$= (\beta-\alpha)\Pr[Y_{(i)} \geq \beta].$$

Combining the two equations above, we see that $\Pr[Y_{(i)} \geq \beta] \leq \frac{i}{n}$. Similarly we may take $S' = \{n-i+1, \cdots, n\}$. Strong Proportionality then implies

$$\mathbb{E}[d(x_n, f(x^{\alpha,\beta}))] \leq \frac{n-i}{n}(\beta-\alpha).$$

However, we also see that

$$\mathbb{E}[d(x_n, f(x^{\alpha,\beta}))] \geq (\beta-\alpha)\Pr[Y_{(i)} \leq \alpha] + 0 \cdot \Pr[Y_{(i)} \geq \beta]$$
$$= (\beta-\alpha)\Pr[Y_{(i)} \leq \alpha].$$

Combining the two equations above, we have $\Pr[Y_{(i)} \leq \alpha] \leq \frac{n-i}{n}$. Thus we see that the probability distribution of $Y_{(i)}$ must satisfy

$$\begin{cases} \Pr[Y_{(i)} \leq \alpha] \leq \frac{n-i}{n}, \\ \Pr[Y_{(i)} \geq \beta] \leq \frac{i}{n}, \end{cases} \qquad \text{for any} \quad 0 \leq \alpha < \beta \leq 1. \tag{1}$$

Letting $\beta = 1$, we see that $\Pr[Y_{(i)} = 1] \leq \frac{i}{n}$, and for any $\alpha < 1$ we have $\Pr[Y_{(i)} > \alpha] \geq \frac{i}{n}$ by condition (1). Taking the limit $\alpha \to 1$ shows $\Pr[Y_{(i)} = 1] = \frac{i}{n}$.

By a similar argument, by setting $\alpha = 0$ we see that $\Pr[Y_{(i)} = 0] \leq \frac{n-i}{n}$ and for any $\beta > 0$ we have $\Pr[Y_{(i)} < \beta] \geq \frac{n-i}{n}$. Taking the limit $\beta \to 0$ implies $\Pr[Y_{(i)} = 0] = \frac{n-i}{n}$ as desired.

$\square$

By Claim 1, we know that $Y_i \in \{0, 1\}$ for each $i \in \{1, \cdots, n-1\}$. Thus the mechanism output is a distribution over $x_1, \cdots, x_n$. Since $f$ is a Phantom mechanism with $Y_{(i)} \in \{0, 1\}$, we see that for

any $k \in \{1, \cdots, n\}$,

$$\Pr[f(x) = \text{rank}^k(x)] = \Pr[f(x) = \text{med}(\underbrace{0, \cdots, 0}_{n-k}, \underbrace{1, \cdots, 1}_{k-1}, x_1, \cdots, x_n)]$$
$$= \Pr[Y_{(n-k)} = 0, \ Y_{(n-k+1)} = 1]$$
$$= \Pr[Y_{(n-k)} = 0] - \Pr[Y_{(n-k+1)} = 0]$$
$$= \frac{n - (n-k)}{n} - \frac{n - (n-k+1)}{n}$$
$$= \frac{1}{n}.$$

Here the third equality follows from the fact that for any $i \in \{1, \cdots, n\}$, we have $\Pr[Y_{(i)} = 0, Y_{(i+1)} = 1] + \Pr[Y_{(i)} = 0, Y_{(i+1)} = 0] = \Pr[Y_{(i)} = 0]$ and $\Pr[Y_{(i)} = 0, Y_{(i+1)} = 0] = \Pr[Y_{(i+1)} = 0]$. The fourth equality follows by Claim 1.

Thus we see that $f$ is equivalent to running the $\text{rank}^k$ mechanism for each $k \in \{1 \cdots, n\}$ with probability $\frac{1}{n}$. Hence $f$ is the Random Rank mechanism. $\qquad \square$

We remark that the Random Rank mechanism satisfies Strong Proportionality only in expectation, and that the guarantee does not hold for each individual deterministic mechanism in the ex-post sense. However, this is unavoidable since no deterministic mechanism satisfies strategyproofness along with Strong Proportionality by Proposition 1.

We now introduce the auxiliary lemma which was used in the proof of Theorem 3.

**Lemma 1.** *If a mechanism is universally anonymous, universally truthful and Proportional in expectation, then it is also ex-post efficient.*

*Proof.* Let $f$ be an arbitrary mechanism that satisfies universal anonymity, universal truthfulness, and Proportionality in expectation. By Theorem 1, we know that deterministic anonymous and strategyproof mechanisms are Phantom mechanisms. Therefore $f$ must be a probability distribution over Phantom mechanisms. Thus $f$ can be written as $f(x) = \text{med}(Y_1, \cdots, Y_{n+1}, x_1, \cdots, x_n)$, where $Y_i$'s are random variables corresponding to the phantom locations. We also denote the order statistics $Y_{(1)} = \min(Y_1, \cdots, Y_{n+1})$ and $Y_{(n+1)} = \max(Y_1, \cdots, Y_{n+1})$.

**Claim 2:** $\Pr[Y_{(n+1)} = 1] = 1$ and $\Pr[Y_{(1)} = 0] = 1$.

We defer the proof of this claim to the appendix.

By Claim 2 and the property of the median, we have that

$$f(x) = \text{med}(Y_{(1)}, \cdots, Y_{(n+1)}, x_1, \cdots, x_n)$$
$$= \text{med}(0, Y_{(2)}, \cdots, Y_{(n)}, 1, x_1, \cdots, x_n,)$$
$$= \text{med}(Y_{(2)}, \cdots, Y_{(n)}, x_1, \cdots, x_n)$$

almost surely. Thus $f$ can be written as a randomized mechanism over Phantom mechanisms with $n - 1$ Phantoms. Since we know from Theorem 2 that Phantom mechanisms with $n - 1$ Phantoms are efficient, we see that $f$ is ex-post efficient. $\qquad \square$

We find that Theorem 3 also extends to the setting where the domain is $X = \mathbb{R}$. As shown in the appendix, the extension requires a few modifications to the proof.

**Theorem 4.** *A mechanism on the domain $X = \mathbb{R}$ is universally anonymous, universally truthful and Strong Proportional in expectation if and only if it is the Random Rank mechanism.*

We remark that the requirements of Theorem 3 are tight in the sense that if any requirement is weakened or removed, the result fails to hold. As we show in the appendix, if Strong Proportionality in expectation is weakened to Proportionality in expectation, the fairness property can be satisfied by the 'Random Phantom' mechanism, which sets phantoms $y_1 = 0$ and $y_{n+1} = 1$ and draws the remaining phantoms I.I.D. from $U([0, 1])$. This mechanism also satisfies universal truthfulness, ex-post efficiency and universal anonymity.

If universal anonymity is weakened to anonymity in expectation, we have the Random Dictator mechanism, which satisfies universal truthfulness and Strong Proportionality in expectation as it is effectively the same as the Random Rank mechanism.

**Definition 9** (Random Dictator). *The Random Dictator mechanism chooses an agent uniformly at random to be the dictator and runs the Dictator mechanism i.e. places the facility at the dictator's location.*

Random Dictator is not universally anonymous as it is a distribution over dictatorial mechanisms which are not anonymous, but it is anonymous in expectation as each agent's reported location has an equal probability of being selected. Hence an agent that knows the realization of the random coin under Random Dictator may change its labelling to achieve a better outcome - this is not possible under the universally anonymous Random Rank mechanism.

If universal truthfulness is weakened to strategyproofness in expectation, there is a class of mechanisms that have better ex-post fairness guarantees, as we will discuss in the next section.

## 6 Improving Ex-post Fairness

In the previous section, we identified Random Rank as the only mechanism satisfying certain desirable properties. One possible drawback of Random Rank is that while it satisfies Strong Proportionality *ex-ante*, it does not have good ex-post fairness guarantees. We consider a class of mechanisms called AverageOrRandomRank that satisfies Strong Proportionality in expectation.[6]

**Definition 10** (AverageOrRandomRank Mechanism). *The* AverageOrRandomRank$-p$ *mechanism places the facility at the average agent location with $p$ probability, and places the facility at the output of the Random Rank mechanism with $1 - p$ probability.*

When $p \in [0, \frac{1}{2}]$, the class of mechanisms additionally satisfy strategyproofness in expectation.

**Theorem 5.** *The AverageOrRandomRank$-p$ mechanism satisfies Strong Proportionality in expectation and is strategyproof in expectation if and only if $p \in [0, \frac{1}{2}]$.*

**Remark 4.** *This result additionally holds when the domain is $X = \mathbb{R}$, as the proof does not involve the interval endpoints.*

To motivate this class of mechanisms, we refer to the example in Figure 2. When we run Random Rank, the facility is placed at 0 with probability $\frac{2}{3}$, or at $\frac{1}{3}$ with probability $\frac{1}{3}$ - either solution is unfair for the group of agents that do not have the facility at their location. In comparison, the AverageOrRandomRank$-\frac{1}{2}$ mechanism halves the respective probabilities that the facility is placed at 0 or $\frac{1}{3}$, and gives a $\frac{1}{2}$ probability of placing the facility at the average location of $\frac{1}{9}$, which is a fair solution for both groups of agents. To summarize, the AverageOrRandomRank$-\frac{1}{2}$ mechanism has a 50% chance of giving a proportionally fair outcome in the ex-post sense, whilst retaining strategyproofness in expectation.

## 7 Achieving Stronger Ex-ante Fairness

The fairness axiom of Strong Proportionality is a weaker variant of a Strong Proportional Fairness, an axiom proposed by Aziz *et al.* [2021]. Strong Proportional Fairness is defined as follows.

**Definition 11** (Strong Proportional Fairness (SPF)). *A mechanism $f$ satisfies* Strong Proportional Fairness (SPF) *if for any location profile $x$ within range $R$ and subset of agents $S \subseteq N$ within range $r$,*

$$d(x_i, f(x)) \le R\frac{n - |S|}{n} + r \qquad \forall i \in S.$$

This axiom captures fairness concerns between cohesive groups of agents that are near each other but not necessarily at the same location. The motivation behind this axiom is similar to that of proportional representation axioms in social choice which guarantee an appropriate level of representation for a sufficiently large group of agents with "similar" preferences [Sánchez-Fernández *et al.*, 2017; Aziz

---

[6]Feldman and Wilf [2013] look at a similar mechanism, but their focus is on welfare rather than fairness.

*et al.*, 2017]. Incidentally, the Random Rank mechanism satisfies the notion of Strong Proportional Fairness in expectation,[7] leading to the following characterization.

**Theorem 6.** *A mechanism is universally anonymous, universally truthful and SPF in expectation if and only if it is the Random Rank mechanism.*

## 8    Conclusion and Future Directions

In this work, we have proposed a new axiom of proportional fairness for the facility location problem called Strong Proportionality and identified mechanisms that satisfy Strong Proportionality in expectation along with universal anonymity and either universal truthfulness or strategyproofness in expectation.

Our results have many natural extensions. The problem could be extended to multiple dimensions. It would also be interesting to characterize mechanisms that satisfy Strong Proportionality in expectation along with a definition of strategyproofness based on stochastic dominance. Other directions for our work include computing approximation ratios for social cost for mechanisms satisfying proportional fairness properties, extending the problem to multiple facilities and discussing facilities with capacity limits.

## Acknowledgements

This work was partially supported by the ARC Laureate Project FL200100204 on "Trustworthy AI". The authors also thank Barton Lee for valuable comments.

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
