# Appendix

**Haris Aziz**
Computer Science and Engineering
UNSW Sydney
Sydney NSW 2052, Australia
`haris.aziz@unsw.edu.au`

**Alexander Lam**
Computer Science and Engineering
UNSW Sydney
Sydney NSW 2052, Australia
`alexander.lam1@unsw.edu.au`

**Mashbat Suzuki**
Computer Science and Engineering
UNSW Sydney
Sydney NSW 2052, Australia
`mashbat.suzuki@unsw.edu.au`

**Toby Walsh**
Computer Science and Engineering
UNSW Sydney
Sydney NSW 2052, Australia
`t.walsh@unsw.edu.au`

## A   Proof of Proposition 1

**Proposition 1.** *No deterministic and strategyproof mechanism satisfies Strong Proportionality.*

*Proof.* For $n = 2$, consider the location profile with 1 agent at $0$ and 1 agent at $0.5$. Strong Proportionality requires that the facility be placed at $0.25$. Now also consider the location profile with 1 agent at $0$ and 1 agent at $1$. Strong Proportionality requires that the facility be placed at $0.5$. However, this means the agent at $0.5$ in the first location profile can misreport their location as $1$ to have the facility placed at their own location, violating strategyproofness. Thus strategyproofness and Strong Proportionality are incompatible in deterministic mechanisms. □

## B   Proof of Claim 2

**Claim 2:** $\Pr[Y_{(n+1)} = 1] = 1$ and $\Pr[Y_{(1)} = 0] = 1$.

*Proof of Claim 2.* We first show that $\Pr[Y_{(n+1)} = 1] = 1$. Suppose the contrary, that there exists $\beta < 1$ such that $\Pr[Y_{(n+1)} \leq \beta] > 0$. Under the location profile $x = (1, \cdots, 1)$, if $f$ satisfies Proportionality in expectation we must have $\mathbb{E}[d(x_n, f(x))] = 0$. However, this leads to a contradiction since

$$\mathbb{E}[d(x_n, f(x))] \geq (1 - \beta) \Pr[Y_{(n+1)} \leq \beta]$$
$$> 0,$$

where the first inequality follows from the fact that if $Y_{(n+1)} \leq \beta$ then $f(x) \leq \beta$, and thus $d(1, f(x)) \geq (1 - \beta)$.

A similar, symmetric argument can be applied to show that $\Pr[Y_{(1)} = 0] = 1$ holds. □

## C   Extension of Theorem 3 to the real line $\mathbb{R}$

In this section we extend the result of Theorem 3 to the real line. We use the following theorem which characterizes strategyproof and anonymous mechanisms on the real line as Phantom mechanisms.

36th Conference on Neural Information Processing Systems (NeurIPS 2022).

**Theorem 7** (Moulin (1980) ). *A mechanism $f$ on the domain $X = \mathbb{R}$ is strategyproof and anonymous if and only if there exists $(n+1)$ real numbers $y_1, \cdots, y_{n+1} \in \mathbb{R} \cup \{+\infty, -\infty\}$ such that*

$$f(x) = \text{med}(x_1, \cdots, x_n, y_1, \cdots, y_{n+1})$$

We also modify our definition of the Random Rank mechanism. Given a profile of locations $x \in \mathbb{R}^n$, we define

$$\text{rank}^k(x) := \text{med}(\underbrace{-\infty, \ldots, -\infty}_{n-k}, x_1, \ldots, x_n, \underbrace{+\infty, \ldots, +\infty}_{k-1}).$$

The Random Rank mechanism on the real line then chooses $k \in \{1, \cdots, n\}$ uniformly at random and outputs $\text{rank}^k(x)$.

**Theorem 4.** *A mechanism on the domain $X = \mathbb{R}$ is universally anonymous, universally truthful and Strong Proportional in expectation if and only if it is the Random Rank mechanism.*

*Proof.* ( $\implies$ ) By Theorem 7 we know that $f$ is a probability distribution over Phantom mechanisms. For each $i \in \{1, \cdots, n+1\}$, denote $Y_i$ as the random variable corresponding to the location of the $i$'th Phantom. Also denote $Y_{(i)}$ as the random variable corresponding to the $i$'th order statistic.

**Claim 3:** $\Pr[Y_{(n+1)} = +\infty] = 1$ and $\Pr[Y_{(1)} = -\infty] = 1$.

*Proof of Claim 3.* Suppose on the contrary that there exists $\lambda \in \mathbb{R}$ such that $\Pr[Y_{(n+1)} \leq \lambda] > 0$. Consider a location profile $x = (2\lambda, \cdots, 2\lambda)$. If $f$ satisfies Strong Proportionality in expectation then we have $\mathbb{E}[d(x_1, f(x))] = 0$. However, this contradicts the following

$$\mathbb{E}[d(x_1, f(x))] \geq |2\lambda - \lambda| \Pr[Y_{(n+1)} \leq \lambda]$$
$$> 0,$$

where the inequality follows since if $Y_{(n+1)} \leq \lambda$ then $f(x) \leq \lambda$, and thus $d(x_1, f(x)) \geq |\lambda|$.

A similar, symmetric argument can be used to obtain $\Pr[Y_{(1)} = -\infty] = 1$. $\qquad\square$

By Claim 3 we see that only $n - 1$ Phantoms are necessary since

$$f(x) = \text{med}(-\infty, Y_{(2)}, \cdots, Y_{(n)}, x_1, \cdots, x_n, +\infty)$$
$$= \text{med}(Y_{(2)}, \cdots, Y_{(n)}, x_1, \cdots, x_n)$$

For notational convenience, we relabel the remaining $n - 1$ Phantoms such that

$$f(x) = \text{med}(Y_{(1)}, \cdots, Y_{(n-1)}, x_1, \cdots, x_n).$$

**Claim 4:** $\Pr[Y_{(i)} = +\infty] = \frac{i}{n}$ and $\Pr[Y_{(i)} = -\infty] = \frac{n-i}{n}$ for each $i \in \{1, \cdots, n-1\}$.

*Proof of Claim 4.* Using the arguments presented in Claim 1, we see that Strong Proportionality implies

$$\begin{cases} \Pr[Y_{(i)} \leq \alpha] \leq \frac{n-i}{n}, \\ \Pr[Y_{(i)} \geq \beta] \leq \frac{i}{n}, \end{cases} \qquad \text{for any} \quad \alpha < \beta, \quad \alpha, \beta \in \mathbb{R}. \qquad (1)$$

From above we see that indeed $\Pr[Y_{(i)} = +\infty] = \frac{i}{n}$ and $\Pr[Y_{(i)} = -\infty] = \frac{n-i}{n}$. $\qquad\square$

By Claim 4, we see that $Y_{(i)} \in \{-\infty, +\infty\}$ for each $i \in \{1, \cdots, n-1\}$ and furthermore,

$$\Pr[f(x) = \text{med}(\underbrace{-\infty, \cdots, -\infty}_{n-k}, \underbrace{+\infty, \cdots, +\infty}_{k-1}, x_1, \cdots, x_n)]$$
$$= \Pr[Y_{(n-k)} = -\infty, \; Y_{(n-k+1)} = +\infty]$$
$$= \Pr[Y_{(n-k)} = -\infty] - \Pr[Y_{(n-k+1)} = -\infty]$$
$$= \frac{n - (n-k)}{n} - \frac{n - (n-k+1)}{n}$$
$$= \frac{1}{n}.$$

The third equality follows from the fact that for any $i \in \{1, \cdots, n\}$, we have $\Pr[Y_{(i)} = -\infty, Y_{(i+1)} = +\infty] + \Pr[Y_{(i)} = -\infty, Y_{(i+1)} = -\infty] = \Pr[Y_{(i)} = -\infty]$ and $\Pr[Y_{(i)} = -\infty, Y_{(i+1)} = -\infty] = \Pr[Y_{(i+1)} = -\infty]$.

Hence we see that $f$ is equivalent to running $\text{rank}^k$ mechanism for each $k \in \{1 \cdots, n\}$ with probability $\frac{1}{n}$. Thus indeed $f$ is the Random Rank mechanism.

( $\impliedby$ ) Similar to the case when $X = [0, 1]$, the Random Rank mechanism is universally anonymous and universally truthful when the domain is $X = \mathbb{R}$ as each realization of the mechanism, $\text{rank}^k$, is strategyproof and anonymous by Theorem 7. The proof that Random Rank satisfies Strong Proportionality in expectation is identical that in the proof of Theorem 3. $\square$

**Remark 1.** *Note that the Phantoms are random variables on the extended real line $\mathbb{R} \cup \{+\infty, -\infty\}$, and thus a random variable $Y$ may satisfy $\Pr[Y = +\infty] > 0$. This is in contrast to random variables defined on $\mathbb{R}$ in which every random variable $Y$ must satisfy $\lim_{N \to \infty} \Pr[Y \geq N] = 0$.*

# D    I.I.D. Phantom Mechanisms

**Definition 1** (I.I.D Phantom Mechanism). *A mechanism is an* I.I.D Phantom *mechanism if it is a Phantom mechanism with $y_1 = 0$, $y_{n+1} = 1$ and the remaining phantoms $y_1, \ldots, y_{n-1}$ are drawn I.I.D according to some distribution $D$ on $[0, 1]$*

The I.I.D Phantom mechanisms are universally truthful, ex-post efficient and universally anonymous, as they only give positive support to instances of deterministic Phantom mechanisms with $y_1 = 0$ and $y_{n+1} = 1$, which by Theorem 2 are strategyproof, efficient and anonymous. If the expected values of the Phantom distribution's order statistics are uniformly spaced on $[0, 1]$, then the mechanism also satisfies Proportionality in expectation.

**Theorem 8.** *An I.I.D Phantom mechanism with distribution $D$ satisfies Proportionality in expectation if and only if the order statistics $D_{(i)}$ have expected value $\mathbb{E}[D_{(i)}] = \frac{i}{n}$ for each $i \in \{1, \cdots, n-1\}$.*

*Proof.* ( $\implies$ ) Fix any $i \in \{1, \cdots, n-1\}$. Consider a location profile $x = (\underbrace{0, \cdots, 0}_{n-i}, \underbrace{1, \cdots, 1}_{i})$ and let $S^0$ be the set of agents located at 0, thus $|S^0| = n - i$. Denote $D_{(i)}$ as the random variable corresponding to the location of the $i$'th order statistic of the Phantoms. Since our mechanism is a Phantom mechanism the output location of the mechanism is distributed as $D_{(i)}$. Thus for any $i \in S^0$ we have

$$\begin{aligned} \mathbb{E}[D_{(i)}] &= \mathbb{E}[d(0, f(x))] \\ &= \mathbb{E}[d(x_i, f(x))] \\ &\leq \frac{n - |S^0|}{n} \\ &= \frac{i}{n} \end{aligned}$$

where the second last equality holds since $f$ satisfies Proportionality in expectation. Similarly let $S^1$ be the set of agents located at 1, and thus $|S^1| = i$. For $j \in S^1$, by proportionality in expectation we see that

$$\begin{aligned} \mathbb{E}[d(x_j, f(x))] &= \mathbb{E}[d(1, f(x))] \\ &\leq \frac{n - |S^1|}{n} \\ &= \frac{n - i}{n} \end{aligned}$$

Since $\mathbb{E}[d(1, f(x))] = 1 - \mathbb{E}[D_{(i)}]$, by rearranging above we see that $\mathbb{E}[D_{(i)}] \geq \frac{i}{n}$. Hence indeed $\mathbb{E}[D_{(i)}] = \frac{i}{n}$ for each $i \in \{1, \cdots, n-1\}$ as needed to show.

( $\impliedby$ ) For any $x \in \{0, 1\}^n$, let $S^0$ be the set of agents located at 0 and $S^1$ be the set of agents located at 1. Let $|S^0| = k$ and $|S^1| = n - k$, the location of the facility is distributed according

to $D_{(n-k)}$. Hence for any $i \in S^0$, we have $\mathbb{E}[d(x_i, f(x))] = \mathbb{E}[D_{(n-k)}] = \frac{n-|S^0|}{n}$. Similarly for $j \in S^1$, we have $\mathbb{E}[d(x_j, f(x))] = 1 - \mathbb{E}[D_{(n-k)}] = 1 - \frac{n-k}{n} = \frac{n-|S^1|}{n}$ as desired. $\qquad\square$

By Theorem 8, we know that the Random Phantom mechanism is Proportional in expectation.

# E   Proof of Theorem 5

**Theorem 5.** *The AverageOrRandomRank$-p$ mechanism satisfies Strong Proportionality in expectation and is strategyproof in expectation if and only if $p \in [0, \frac{1}{2}]$.*

*Proof.* We first show that the mechanism is Strong Proportional in expectation. Consider any location profile $x \in \{\alpha, \beta\}^n$, and let $S_\alpha$ denote the set of agents at $\alpha$ and $S_\beta = N \backslash S$ denote the set of agents at $\beta$. The AverageOrRandomRank$-p$ mechanism places the facility at:

- $\alpha$ with probability $(1-p)\frac{|S_\alpha|}{n}$,

- at $\beta$ with probability $(1-p)\frac{|S_\beta|}{n}$,

- and at $\frac{|S_\alpha|\alpha + |S_\beta|\beta}{n}$ with probability $p$.

For all $i \in S_\alpha$, we have

$$\mathbb{E}[d(x_i, f_{RR}(x))] = (1-p)\frac{|S_\beta|}{n}(\beta - \alpha) + p\left(\frac{|S_\alpha|\alpha + |S_\beta|\beta}{n} - \alpha\right)$$
$$= \frac{|S_\beta|}{n}\beta - \alpha(1-p)\frac{|S_\beta|}{n} + p\alpha\frac{|S_\alpha| - n}{n}$$
$$= \frac{|S_\beta|}{n}(\beta - \alpha) = \frac{n - |S_\alpha|}{n}(\beta - \alpha),$$

and for all $j \in S_\beta$, we have

$$\mathbb{E}[d(x_j, f_{RR}(x))] = (1-p)\frac{|S_\alpha|}{n}(\beta - \alpha) + p\left(\beta - \frac{|S_\alpha|\alpha + |S_\beta|\beta}{n}\right)$$
$$= -\frac{|S_\alpha|}{n}\alpha + \beta(1-p)\frac{|S_\alpha|}{n} + p\beta\frac{n - |S_\beta|}{n}$$
$$= \frac{|S_\alpha|}{n}(\beta - \alpha) = \frac{n - |S_\beta|}{n}(\beta - \alpha).$$

Hence, AverageOrRandomRank$-p$ satisfies Strong Proportionality in expectation.

We now show that the mechanism is strategyproof in expectation. Suppose an agent at $x_i$ deviates by distance $d$ to attain a better expected distance. Its expected cost is reduced by $\frac{dp}{n}$ from the average location moving closer, but is also increased by $\frac{d(1-p)}{n}$ from its reported location moving away. For strategyproofness we require that $\frac{d(1-p)}{n} \geq \frac{dp}{n}$, which is satisfied for $p \in [0, \frac{1}{2}]$. Furthermore, it is easy to see that if $p > \frac{1}{2}$, an agent can improve its expected distance from the facility by misreporting its location. $\qquad\square$

# F   Proof of Theorem 6

**Theorem 6.** *A mechanism is universally anonymous, universally truthful and SPF in expectation if and only if it is the Random Rank mechanism.*

*Proof.* Since SPF implies Strong Proportionality, by Theorem 3 it suffices to prove Random Rank satisfies SPF. Consider any location profile $x$ within range $R$ and subset of agents $S \subseteq N$ within

range $r$. Denote $X_S$ as the event that Random Rank places the facility at an agent in $S$. Then for any $i \in S$, we have

$$\mathbb{E}[d(x_i, f(x))] \leq R(1 - \Pr[X_S]) + r \Pr[X_S]$$
$$\leq R \left( \frac{n - |S|}{n} \right) + r \frac{|S|}{n}$$
$$\leq R \left( \frac{n - |S|}{n} \right) + r.$$

$\square$