# OpenReview forum: "Random Rank: The One and Only Strategyproof and Proportionally Fair Randomized Facility Location Mechanism"
_NeurIPS.cc/2022/Conference — NeurIPS 2022 Accept_

### Official Review · Reviewer_BKnC · 2022-07-08

**Rating:** 6
**Confidence:** 3
**Soundness:** 4 excellent
**Presentation:** 4 excellent
**Contribution:** 3 good

**Summary:**

The paper studies mechanisms for facility location games that are both truthful and fair.  In particular, the authors consider a fairness notion called Strong Proportionality, which roughly says when there are two groups of agents where each group are located at the same location, the total costs of the two groups should be the same in expectation.  The main result is that the only mechanism that is anonymous (meaning symmetric across agents), truthful, and strongly proportional is Random  Rank, which picks an integer k uniformly at random between 1 and n and outputs the k-th largest location.

**Questions:**

(Also putting some detailed comments here.)

Line 70, "k-facility: the "-" looks like a minus sign.

"AverageOrRandomRank-p": again the "-" looks like a minus sign.

Sec 6: I'm a bit confused by the logical flow here.  Random Rank is only fair in expectation, and the fix is another mechanism that is also only fair in expectation?  Is the idea that the new mechanism is "more fair" (although still not "exactly fair") in the ex-post sense?

**Limitations:**

Overall I'm satisfied (see detailed comments above for some minor suggestions).

**Strengths And Weaknesses:**

Strengths

The paper is well written and polished.  Fairness in facility location games is a meaningful problem.  The main result of the paper is clean and nice, and the proof is cute and nontrivial.  Conceptually, I mostly view the result as a negative message saying proportionality is hard to achieve in facilitty location games, which is of course a meaningful message.


Weaknesses

My biggest concern is regarding the proportionality axiom itself.  While the axiom definitely makes sense, it's not clear to me it is the only reasonable notion of fairness in the context of facility location games.  To this end, I'd appreciate more discussion on other notions of fairness (if they exist) and why they are not as suitable for the purposes of this paper.  A less important thing is I'm curious about the possibility of a characterization for truthful-in-expectation mechanisms.  There presumably one would need some new techniques, since the characterization by Massó and Moreno De Barreda no longer applies.

---

> ### Author Response · Authors · 2022-07-30
> **Response to Reviewer 4**
>
> Thank you for your kind and valuable comments, and for your suggestion to discuss other notions of fairness.
>
> 1. “I'd appreciate more discussion on other notions of fairness (if they exist) and why they are not as suitable for the purposes of this paper.”
>
> Our work focuses on capturing proportional fairness concerns for groups of agents at the same location, thus our axiom of Strong Proportionality which leads to a unique characterization is most suitable for our setting. While our setting and axioms are most similar to those discussed in the work by Aziz et al (2021), the other existing fairness notions in the literature do not precisely capture the concerns that we are aiming for.
>
> Indeed, there are various other notions of fairness that have been proposed in facility location games. As we have mentioned in the Related Work section, the work by Aziz et al. (2021) provides a comprehensive overview of some distance-based proportional fairness notions in facility location games, and another related paper by Zhou et al. (2021) looks at group-fairness objectives. In addition, there is the egalitarian fairness notion of ‘maximum cost’, in which we aim to minimize the distance of the worst-off agent from the facility. The classic paper which discusses the approximation of optimal maximum cost (and optimal total cost) by strategyproof mechanisms is by Procaccia and Tennenholtz (2013).
>
> Thank you again for your suggestion. We will add further discussions on other fairness notions to the Related Work section of the paper.
>
> 2. “Random Rank is only fair in expectation, and the fix is another mechanism that is also only fair in expectation? Is the idea that the new mechanism is "more fair" (although still not "exactly fair") in the ex-post sense?”
>
> Yes, you are correct in saying that the AverageOrRandomRank-p mechanism is only fair in expectation, and that it leads to ‘better’ ex-post outcomes. Taking an axiomatic viewpoint and setting p=1/2, the AverageOrRandomRank-½ mechanism satisfies Strong Proportionality in expectation and is not guaranteed to satisfy it ex-post. However, we can see that there is now a 50% chance that the outcome is guaranteed to be proportionally fair in the ex-post sense, and that the compromise is universal truthfulness is weakened to strategyproofness in expectation.
>
> Thank you for asking for further clarification on this point. We will add this explanation in the revised version of the paper.

---

> > ### Comment · Reviewer_BKnC · 2022-08-08
> > **Thanks for the response**
> >
> > The response does answer my questions.

---

### Official Review · Reviewer_vt86 · 2022-07-10

**Rating:** 4
**Confidence:** 4
**Soundness:** 4 excellent
**Presentation:** 4 excellent
**Contribution:** 2 fair

**Summary:**

This work considers the classical facility location problem with a new fairness constraint, "strong proportionality." The authors show that no deterministic strategy can be both strategyproof and fair, and further leverage known characterizations of strategyproof mechanisms in this setting to identify ``Random Rank" as the unique (randomized) mechanism that is both strategyproof and fair.

**Questions:**

---Again, I would like the thank the authors for the clear and respectful exposition.

---To elaborate on the ``weaknesses" above, how surprising is the main result? While I am not an expert on this literature, the strong characterization of strategyproof mechanisms seems to make the result somewhat unsurprising, at least naively. This rigidity further manifests in the limited setting of the problem to $[0,1]$ or $\mathbb{R}$.

---One could even argue that the main takeaway of the main result is that it shows that strong proportionality is itself not a sufficiently interesting fairness notion, especially given that it seems to have been first introduced in the present paper.

---That said, I do wish to emphasize that the arguments in this work are quite nice in their own right. However, it seems to me that the results in the present paper would be a great complement to a more thorough investigation of strong proportionality or other related fairness notions, as seems to have been done for proportionality in Aziz, et al (2021).

---The definition of randomized mechanism (Footnote 3) seems slightly ambiguous; it seems like the probability distribution over deterministic algorithms should itself be a function of the input to accommodate the example in the footnote, but this isn't clear in the definition.

**Limitations:**

There do not appear to be obvious negative societal impacts. The authors do discuss several possible avenues for improving the current results.

**Strengths And Weaknesses:**

Strengths: This paper succeeds in providing a complete characterization of strategyproof mechanisms that satisfy their notion of strong proportionality. The paper is quite well-written and respectful to the reader.

Weaknesses: My overall impression is that the results and techniques in this work may be somewhat limited in scope; a more thorough investigation of the ramifications of strong proportionality would make for a very strong paper.

---

> ### Author Response · Authors · 2022-07-30
> **Response to Reviewer 3**
>
> We thank the reviewer for their comments.
>
> In response to the suggestion that " more thorough investigation of the ramifications of Strong Proportionality would make for a very strong paper", we would like to emphasize that we provided a  clear ramification of strong proportionality via the following results:
>
> (1) Strong Proportionality is incompatible with strategyproofness and deterministic mechanisms
>
> (2) Strong Proportionality leads to a unique mechanism as long as we additionally require universal truthfulness and universal anonymity.
>
> 1. “To elaborate on the ``weaknesses" above, how surprising is the main result?”
>
> Our main result is surprising because we have defined three basic desirable properties which we expect any “fair” and truthful mechanism to satisfy, and out of the very large space of randomized mechanisms, there is only one mechanism that satisfies these properties.
>
> 2. “This rigidity further manifests in the limited setting of the problem to [0,1] or R.”
>
> Although our setting is limited to the one-dimensional domain of [0,1] or R, this is sufficient to prove the unique characterization of our main result. Our result implies that any mechanism for higher dimensions will have to coincide with
> RandomRank for uni-dimensional problem instances  if we require the axioms considered in our characterization.
>
> Furthermore, the one dimensional domain is the standard ‘starting point’ for almost every paper in facility location mechanism design (see, e.g. Moulin (1980), Border and Jordan (1983), Procaccia and Tennenholtz (2013)), as the setting itself is readily applicable to settings in participatory budgeting, scheduling and single-peaked preferences, and it can be naturally extended to graphs or higher dimensions in future work.
>
> 3. “One could even argue that the main takeaway of the main result is that it shows that Strong Proportionality is itself not a sufficiently interesting fairness notion, especially given that it seems to have been first introduced in the present paper.”
>
> Although Strong Proportionality was introduced in our paper, it is stronger than the existing notions of proportional fairness which have been used in the literature (Freeman et al. 2019, Aziz et al. 2021). Furthermore, in Section 7, we show that the unique characterization of our main result can be extended so that the RandomRank mechanism satisfies SPF, a pre-existing axiom defined by Aziz et al. (2021).
>
> 4. “The definition of randomized mechanism (Footnote 3) seems slightly ambiguous; it seems like the probability distribution over deterministic algorithms should itself be a function of the input to accommodate the example in the footnote, but this isn't clear in the definition.”
>
> Our definition of randomized mechanisms is correct and consistent with the literature (see, e.g. Assadi and Singla (2019) and Dobzinski and Dughmi (2013)). We will modify footnote 3 to align with this definition to avoid any ambiguity.

---

### Official Review · Reviewer_EUGM · 2022-07-11

**Rating:** 6
**Confidence:** 4
**Soundness:** 4 excellent
**Presentation:** 4 excellent
**Contribution:** 2 fair

**Summary:**

This paper proposes a new concept Strong Proportionality, which means in the case that there are only two groups of agents at two locations, then both groups incur the same total cost. This is a stronger version of the Proportionality property in the participatory budgeting literature. The authors show that there is no deterministic strategyproof mechanism satisfying this property, and designed an algorithm, Random Rank, that satisfies Strong Proportionality in expectation. They also show that Random Rank is the only randomized mechanism that achieves universal truthfulness, universal anonymity, and Strong Proportionality in expectation. Finally, they show that if universal truthfulness could be weakened to truthfulness in expectation, then there is a mechanism that could have a fairer ex-post fairness guarantee.


**Questions:**

Have you considered generalizing the setting to multiple groups in 1 dimension? What makes it much more difficult than the two groups case?

**Limitations:**

The authors mentioned that one future direction would be to extend the problem to multiple dimensions, but even in the one dimension case, it would be nice to have a more general case than only two groups of agents.

**Strengths And Weaknesses:**

* Originality: This paper is closely related to research on facility location and fairness in collective decision making, especially the Proportionality concept in participatory budgeting. The Strong Proportionality concept is naturally motivated. The proofs are nicely built on previous results about the Phantom mechanism. Related work is properly cited.
* Quality: The notations are well defined. The theorems / lemmas are cleanly presented and proved.
* Clarity: This paper is well written and organized. A minor comment: Figure 1 and Table 1 are not referred to in the main paper.
* Significance: The authors have a nice summary of randomized social choice and facility location mechanisms in Table 1, and show how Random Rank is the only algorithm that satisfies all the properties listed. This result is interesting and could inspire new research directions in this area. However, I feel the setting that there are only two groups of agents at two locations is limited.

---

> ### Author Response · Authors · 2022-07-30
> **Response to Reviewer 2**
>
> Thank you for your kind and insightful comments, and for noticing that Figure 1 and Table 1 are not referenced in the main paper. We will add brief sentences in the main text referencing them.
>
> 1.“Have you considered generalizing the setting to multiple groups in 1 dimension? What makes it much more difficult than the two groups case?”
>
> We would like to clarify the reasoning behind restricting the setting to two groups of agents. In our setting, we have not necessarily restricted to two groups of agents in the sense that our mechanisms only apply to this specific case. Rather, what we show is that restricting to only two groups and defining the respective axiom of Strong Proportionality is sufficient to provide a unique characterization of anonymous, strategyproof and proportionally fair mechanisms. Our mechanisms can still be applied to location profiles where there are any number of groups, and as we discuss in Section 7, they are still ‘proportionally fair’ under an axiom that captures fairness concerns for any number of cohesive groups of agents.

---

> > ### Comment · Reviewer_EUGM · 2022-08-08
> > **Thank you for the response.**
> >
> > Thank you for the response!

---

### Official Review · Reviewer_HQVm · 2022-07-11

**Rating:** 7
**Confidence:** 3
**Soundness:** 3 good
**Presentation:** 4 excellent
**Contribution:** 3 good

**Summary:**

In this paper, the author proposes a new concept called Strong Proportionality as a new fairness measurement. The author shows that Strong Proportionality is not achievable using deterministic and strategyproof mechanisms and proposes a randomized mechanism called Random Rank which can achieve not only Strong Proportionality but also other efficiency concepts like universal truthfulness etc.

**Questions:**

I have a few questions:

- In the actual deployment, will the decision-maker picks and fix a deterministic mechanism (choose from the randomized mechanisms?) If that's the case, the theoretical guarantee holds only in expectation over the choice of the deterministic mechanism, which does not hold for every single mechanism. The paper could be stronger if the author could say something like "w.h.p. over the choice of the mechanisms, the deployed mechanism also satisfies strong proportionality" etc.

- Can the authors comment on how the mechanism scales up to a higher dimensional setting? Will there be any curse of dimensionality problem?

**Strengths And Weaknesses:**

Overall, the paper contributes useful concepts, and the definitions and mathematical arguments are clearly written. The impossibility results show us the limitation of deterministic mechanisms and thus the randomized mechanisms are naturally motivated. The paper gaps between the economic and machine learning literature and is a valuable contribution to the community.

---

> ### Author Response · Authors · 2022-07-30
> **Response to Reviewer 1**
>
> Thank you for your kind and valuable comments, and for your interest in our paper.
>
> 1.“In the actual deployment, will the decision-maker picks and fix a deterministic mechanism (choose from the randomized mechanisms?) If that's the case, the theoretical guarantee holds only in expectation over the choice of the deterministic mechanism, which does not hold for every single mechanism.”
>
> Since randomized mechanisms are defined as probability distributions over deterministic mechanisms, each realization of our randomized mechanism is a deterministic mechanism. Indeed, in the actual deployment of a randomized mechanism, the mechanism designer will select and run a deterministic mechanism at random. Hence you are correct in that our mechanisms satisfy the Strong Proportionality property only in expectation, and that the guarantee does not hold for each individual deterministic mechanism in the ex-post sense. However this is unavoidable since no deterministic mechanism satisfies strategyproofness along with Strong Proportionality by Proposition 1. We will add this explanation to the paper.
>
> In section 6, we relax universal truthfulness to strategyproofness in expectation to achieve better ex-post guarantees via the AverageOrRandomRank-1/2 mechanism. Under this randomized mechanism, there is a ½  probability that the deployed mechanism satisfies Strong Proportionality.
>
> 2. “Can the authors comment on how the mechanism scales up to a higher dimensional setting? Will there be any curse of dimensionality problem?”
>
> Thank you for this interesting question. Firstly, we remark that in higher dimensions, (dimension greater or equal to three) no deterministic mechanism is strategyproof, anonymous and efficient by a result of Peters et al [1]. The consequence of this result is that there is no analogue of the Random Rank mechanism for higher dimensions. As each deployed deterministic mechanism is strategyproof and efficient, it is impossible for the mechanism to also be anonymous. In other words, we cannot find a way to order the agents such that anonymity is preserved for each individual deterministic mechanism. Thus when there are three or more dimensions, the desirable property of universal anonymity cannot be achieved by scaling up our mechanism. However, we can achieve the weaker property of anonymity in expectation (along with universal truthfulness and Strong Proportionality in expectation) by instead using the random dictatorship mechanism.
>
> [1] Peters, H., van der Stel, H., & Storcken, T. (1992). Pareto optimality, anonymity, and strategy-proofness in location problems. International Journal of Game Theory, 21(3), 221-235.

---

### Meta-Review · Area_Chair_H8N3 · 2022-08-26

**Recommendation:** Accept
**Confidence:** Certain

**Metareview:**

Executive summary:

This paper considers the facility location problem. The authors define and explore a natural new fairness notion called "strong proportionality" (Definition 5). They show that no mechanism can be universally truthful and universally strongly proportional; at the same time they show that the randomized RandomRank mechanism they devise is the unique mechanism satisfying universal anonymity, universal truthfulness, and strong proportionality in expectation.

Discussion:

There is considerable support for this paper (one accepts, two weak accepts, and only one borderline reject). All reviewers appreciate the model and the new fairness notion, and that it uniquely characterizes (a natural) randomized mechanism.

The only slightly negative reviewer still liked the paper, but was worried the fairness notion may be limited to the specific problem and wasn't explored more generally.

Accept.

**Award:**

No

---

### Decision · Program_Chairs · 2022-09-14

Accept